# CoRNStack: High-Quality Contrastive Data for Better Code Retrieval and Reranking

**Tarun Suresh**[1,3*†]   **Revanth Gangi Reddy**[1*]   **Yifei Xu**[1,2]   **Zach Nussbaum**[3]
**Andriy Mulyar**[3]   **Brandon Duderstadt**[3]   **Heng Ji**[1]
[1]University of Illinois Urbana-Champaign   [2]Lapis Labs   [3]Nomic AI
{tsuresh3,revanth3,yifeix5,hengji}@illinois.edu

## Abstract

Effective code retrieval plays a crucial role in advancing code generation, bug fixing, and software maintenance, particularly as software systems increase in complexity. While current code embedding models have demonstrated promise in retrieving code snippets for small-scale, well-defined tasks, they often underperform in more demanding real-world applications such as bug localization within GitHub repositories. We hypothesize that a key issue is their reliance on noisy and inconsistent datasets for training, which impedes their ability to generalize to more complex retrieval scenarios. To address these limitations, we introduce CoRNStack[1], a large-scale, high-quality contrastive training dataset for code that spans multiple programming languages. This dataset is curated using consistency filtering to eliminate noisy positives and is further enriched with mined hard negatives, thereby facilitating more effective learning. We demonstrate that contrastive training of embedding models using CoRNStack leads to state-of-the-art performance across a variety of code retrieval tasks. Furthermore, the dataset can be leveraged for training code reranking models, a largely underexplored area compared to text reranking. Our finetuned code reranking model significantly improves the ranking quality over the retrieved results. Finally, by employing our code retriever and reranker together, we demonstrate significant improvements in function localization for GitHub issues, an important component of real-world software development.

## 1 Introduction

The rapid advancement of software development has led to an increased reliance on automated tools for code generation (Chen et al., 2021; Li et al., 2022b; Nijkamp et al.; Ugare et al., 2024b; Banerjee et al., 2025; Ugare et al., 2024a; Suresh et al., 2025). As codebases grow in both size and complexity, the ability to efficiently search for and retrieve relevant code snippets is important. Code retrieval is crucial for advancing Retrieval-Augmented Code Generation (RACG) (Wang et al., 2024b) with large language models (LLMs), where providing contextual examples significantly improves the relevance and accuracy of generated code. Effective code retrieval facilitates bug identification, enables the reuse of existing solutions, and minimizes redundancy, accelerating the overall development process. Code embedding models (Li et al., 2022a; Wang et al., 2023b; Zhang et al., 2024) have gained traction for their ability to encode the semantic and syntactic properties of code into dense vector representations and help retrieve relevant snippets with high precision. However, these approaches have not demonstrated substantial success in real-world applications, particularly in complex tasks like resolving GitHub issues as evaluated by benchmarks like SWE-Bench (Jimenez et al., 2024).

We hypothesize that code embedding models often suffer from suboptimal training procedures. Most state-of-the-art code embedding models rely on contrastive learning (Li et al., 2022a), which is a powerful technique for learning representations by reducing the distance between similar code snippets while maximizing the distance between dissimilar ones. Yet, existing methods predominantly

---

*Equal Contribution.

†Work done during an internship at Nomic AI.

[1]Dataset, code and models are available at: https://github.com/gangiswag/cornstack

fine-tune these models on noisy bimodal (text, code) datasets (Husain et al., 2019; Kocetkov et al.) heuristically sourced from open platforms like GitHub. These datasets usually lack curation and consistency filtering mechanisms, leading to significant noise, including irrelevant or incorrectly labeled pairs, which impairs the model's ability to learn robust representations. Further, existing approaches often fail to incorporate such challenging negatives, resulting in embeddings that struggle to capture fine-grained distinctions between similar code snippets. This limitation prevents current models from effectively handling subtle semantic differences, thus compromising their retrieval accuracy in real-world scenarios.

To address these issues, we curate CORN-STACK[2], a large-scale dataset of high-quality (text, code) pairs based on The Stack V2 (Lozhkov et al., 2024), refined through consistency filtering, and supplemented with mined hard negatives for effective contrastive learning. To remove noisy (text-positive) pairs, our approach uses a dual consistency filtering process, which ensures the positives are within top-$k$ across the corpus, while having an embedding similarity score greater than a threshold. Further, we incorporate a hard negative mining strategy with softmax-based sampling over a larger collection of negatives to promote diversity, with a curriculum controlling the sampling probability to ensure we progressively increase the difficulty of the sampled negatives across the training process. We demonstrate that contrastive learning using our dataset leads to state-of-the-art code embedding models, outperforming even larger models (Zhang et al., 2024; Wang et al., 2023b) on a variety of code retrieval tasks (Husain et al., 2019; Huang et al., 2021; Lu et al., 2021).

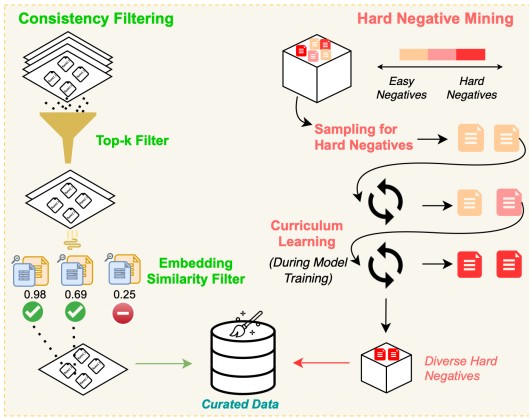

Figure 1: Figure demonstrating the curation methodology for CORNSTACK, with consistency filtering to remove noisy positives in addition to a curriculum-based hard negative mining strategy.

In addition to embedding quality, code retrieval quality can benefit from sophisticated reranking techniques. While reranking has been extensively applied in domains like text retrieval (Zhuang et al., 2023b; Sun et al., 2023) and recommendation systems (Liu et al., 2022; Gao et al., 2024), its application in code retrieval remains largely unexplored. A major challenge for building effective code reranking models is the lack of high-quality data for fine-tuning on contrastive bimodal (text, code) data. In this paper, we show that our dataset, with the curated set of (text, code) pairs that involve positives and negatives, can be leveraged to finetune code generation models (Wang et al., 2023b; Hui et al., 2024) to be state-of-the-art code rerankers.

Finally, using our improved code retriever and reranker together, we show significant improvements in function localization (Xia et al., 2024), a key aspect in addressing real-world software engineering challenges (Jimenez et al., 2024). Function localization refers to the ability to accurately identify the specific functions or code segments that require modification in response to a particular issue, such as a bug report or feature request, especially on platforms like GitHub. Our retriever first identifies a pool of highly relevant code snippets, while the reranker further refines these results, prioritizing the most contextually appropriate functions based on their relevance to the given issue.

Our main contributions can be summarized as follows:

- We introduce CORNSTACK, a large-scale curated high-quality (text, code) pairs dataset, refined through consistency filtering, and supplemented with mined hard negatives for effective contrastive learning.
- We show that code retrievers trained using CORNSTACK have considerably higher performance on a variety of code retrieval benchmarks, with substantial gains over current state-of-the-art code embedding models, while using a considerably smaller encoder.
- We are the first to finetune LLMs as code rerankers. Our 7B code reranker, trained leveraging our contrastive data, considerably improves performance over the retriever.

---

[2]CORNSTACK stands for **Co**nsistency filtering and **R**obust **N**egatives for enriching The **Stack** v2.

- We demonstrate the benefit of improved code retrieval and reranking on function localization, while solving real-world software development problems such as addressing GitHub issues.

## 2    CORNSTACK

The performance of code embedding models is highly contingent on the quality of the large-scale data used for contrastive training, in the form of <query, positive, negatives> triples. Effective contrastive training hinges on satisfying two primary conditions: 1) The positives are highly relevant to the query and not noisy, 2) The negatives are semantically similar to the positives but do not directly address the query, a.k.a *hard* negatives. Heuristically sourcing contrastive examples from large-scale open-source code data, such as the Stackv2 (Lozhkov et al., 2024), can include irrelevant or incorrectly labeled <query, positive> pairs, which impair the models' ability to learn robust and accurate representations. Hence, we introduce a two-step consistency filtering method (in §2.2) that selects positives that are among the top-$k$ in the corpus and have an embed-

| Language | Original | Selected |
|---|---|---|
| Python | 46.1M | 10M |
| JavaScript | 85.5M | 2.3M |
| Java | 113.3M | 3M |
| Go | 9.2M | 2M |
| PHP | 31.8M | 3M |
| Ruby | 10.8M | 0.9M |

Table 1: # instances in CORNSTACK for various languages after filtering.

ding similarity above a certain threshold. Additionally, we employ a hard negative mining strategy (see §2.3) to augment the <query, positive> pairs with negatives, sampled over a larger collection to ensure diversity. We also implement a curriculum that adjusts the sampling probabilities, progressively increasing the difficulty of the negatives throughout the training process. We call this collection of contrastive training examples CORNSTACK, with details in §3 on leveraging this data to finetune code retrievers and rerankers. Table 1 shows the counts of examples from different languages[3] in our contrastive dataset after filtering from the original Stackv2.

### 2.1    DATA SELECTION

We base our dataset on the de-duplicated version of The Stack v2[4], a comprehensive collection of source code in 600+ programming and markup languages. We convert this into bimodal data, i.e. (text, code) pairs, by extracting the docstring of a function as the text, and the corresponding function as the code. Following Zhang et al. (2024), we heuristically filter out pairs when the text is not in English, it is too short which removes URLs, HTML tags, and other bad Unicode characters in the text. To ensure the syntactic correctness of the code data in CoRNStack, following Guo et al. (2021), we used the Tree-sitter parsing toolkit to filter out any codes that cannot be parsed into a syntax tree. However, unlike previous works, we do not filter pairs with $\geq 256$ tokens in the text. By not filtering out these long-text pairs, we aim to improve the ability of the model to generalize well to long sequences in the queries, which are common in repository-level code retrieval tasks. GitHub issues, for instance, often contain long, detailed descriptions of problems or feature requests.

### 2.2    DUAL CONSISTENCY FILTERING

In analyzing the heuristically collected text-code pairs from the Stack-v2, we observe that many docstrings inadequately describe the corresponding code behavior. Additionally, there are cases where the code does not perform the functionality described in the docstring. Such discrepancies can be harmful during training, as they provide a noisy signal for relevance between the natural language descriptions and corresponding code implementations.

To address the data quality limitations, we build upon recent work in consistency filtering from training text embedding models (Wang et al., 2022; Günther et al., 2023). Consistency filtering aims to curate a refined dataset by excluding training pairs with low semantic similarity. In this work, we incorporate *dual* consistency filtering with two criteria. First, for a given query, we ensure that the positive code snippet is among the top-$k$ most semantically similar snippets in the dataset.

---

[3]While we were left with 16M for Java and 7M for Go after filtering, we subsampled for these languages to ensure Python covered 50% of the data since most downstream benchmarks are based on python. These six languages were picked to cover those evaluated in CodeSearchNet Husain et al. (2019).

[4]https://huggingface.co/datasets/bigcode/the-stack-v2-dedup

This top-$k$ retrieval step filters out irrelevant or weakly related code snippets. Next, we apply a secondary filtering stage where pairs with similarity scores below a predefined threshold, $\delta$, are discarded. This guarantees that even highly ranked pairs must surpass a minimal quality threshold, reducing the retention of pairs that are only marginally relevant. The relative and absolute thresholds are determined to balance dataset size and quality, ensuring that only the most consistent pairs are included.

Formally, let $(t_i, c_i)$ denote the heuristically collected (text, code) pair, $T = [t_1, t_2, \ldots, t_n]$ represent the list of texts, and $C = [c_1, c_2, \ldots, c_n]$ represent the corresponding code snippets in the corpus $D$. We use an existing embedding model $N$ to encode both texts and code snippets into vector representations: $T_v = N(T)$ and $C_v = N(C)$. A similarity score matrix $S = T_v \cdot C_v^T$ is computed, where each entry $S_{i,i}$ represents the similarity between text $t_i$ and code $c_i$. For all $(t_i, c_i) \in D$, $(t_i, c_i)$ is included to the curated dataset $D'$ if $S_{ii}$ is in the top-$k$ ranked values of $S[i]$ and $S_{ii} > \delta$, the similarity threshold[5], $k$=2 and $\delta$=0.7.

| Dataset | # Examples | % Correct |
|---------|------------|-----------|
| CosQA | 20k | 63.9 |
| CSN | 2M | 55.8 |
| Stack v2 | 200M | 52.9 |
| Ours | 21M | 77.1 |

Table 2: Evaluation of <query, positive> pair correctness for different code corpora.

To demonstrate the superior quality of pairs in CORNSTACK, we perform an automated evaluation, comparing our filtered dataset against other contrastive code datasets, such as CosQA Huang et al. (2021) and CodeSearchNet (CSN) Husain et al. (2019). Specifically, we prompt Qwen2.5-Coder-7B-Instruct (Hui et al., 2024), an instruction-tuned code generation model to judge whether a code snippet fully answers the corresponding query for 10k randomly sampled pairs from each dataset for 3 seeds. Table 2 shows the mean % correctness evaluated by the LLM, with our large-scale contrastive collection considerably improving in quality over The Stack v2, CodeSearchNet, and CosQA. We show the language-wise % correctness in Table A.2 in the Appendix.

## 2.3 HARD NEGATIVE MINING

Beyond improving the semantic relevance of positive pairs, incorporating challenging negatives is critical to improving the model's ability to distinguish semantically similar instances, as seen in text embedding literature (Wang et al., 2024a; Moreira et al., 2024). Several prior works in code embedding model training attempt to mine hard negatives but face key limitations. CodeSage (Zhang et al., 2024) weights in-batch negatives based on relevance to the query. However, since the set of in-batch negatives is chosen at random, this approach is still limited by the hardness of negatives available within a single batch. CodeT5+ (Wang et al., 2023b) uses a contrastive similarity score to sample negatives from a queue maintained by a momentum encoder (He et al., 2020). However, this approach is restricted by the queue size and introduces high memory overhead. Moreover, both methods risk sampling false negatives, which degrades contrastive learning by introducing noise. In this work, we introduce a hard negative mining strategy that involves sampling negatives from a large pool to promote diversity. Our hard negative mining strategy is divided into 2 stages: 1) an offline stage that leverages the corpus-level similarity score matrix $S$ pre-computed during consistency filtering to filter false-negatives, and 2) an online stage that uses a softmax-based sampling strategy with curriculum to progressively select diverse, challenging negatives during the contrastive fine-tuning.

Given a positive (text, code) pair in the dataset, denoted as $(t_i, c_i^+)$, and let $B_i = \{c_j^-\}_{j=1}^M$ represent the set of hard negatives for text $t_i$, with $S$ being the similarity score matrix between all the text and code snippets in the corpus. To eliminate false negatives, we follow Moreira et al. (2024) to remove any $c_j^-$ which is sufficiently close to $t_i$. Specifically, we filter out any negative $c_j^-$ for which the relevance score $S_{ij}$ exceeds a threshold $\gamma \cdot S_{ii}$, where $S_{ii}$ is the similarity score text $t_i$ and positive code snippet $c_i$. We then cache $S$ for use in the online stage of our negative mining.

---

[5]In our experiments, we use Jina-Code-v2 (Günther et al., 2023) as the proxy embedding model $N$

## 3 CODE RETRIEVER AND RERANKER

Each instance in our curated high-quality CORNSTACK dataset is of the form of the form of <query, positive, negatives> triples, which can be used for contrastive training. Here, we describe how CONTRASTACK is leveraged for finetuning both code retriever and reranker models.

### 3.1 CODERANKEMBED RETRIEVER

We use a bi-encoder architecture (Reimers & Gurevych, 2019) for our retriever, with weights shared between the text and code encoder. Let $(t_i, c_i^+)$ denote a positive (text, code) pair in the dataset and $(h_i, h_i^+)$ be the respective output representations from the last hidden layer of the encoder. For a batch of size $N$, let $H = \{h_i^+\}_{i=1}^N$ denote the code representations from the positive code samples in the batch. Let $H_B = \bigcup_{i=1}^n \{h_{ij}^-\}_{j=1}^M$ denote the set of hard negatives code representations of all text $t_i$ in the batch sampled via an online mining strategy that we describe next.

We use the pre-computed similarity-score matrix $S$ (from §2.3) and for each $t_i$, sample $M$ negatives with probability: $P(c_j^-|t_i) = \exp(S_{ij}/\tau')/(\sum_{m=1}^M \exp(S_{im}/\tau'))$ where $\tau'$ is the temperature parameter[6]. Different from recent work in negative mining for text-embedding models Moreira et al. (2024) that select from top-k negatives, our softmax-based sampling introduces diversity into the selection of negatives, increasing the exploration of the negative sample space while maintaining a high likelihood of selecting challenging negatives. This strategy avoids overfitting to specific hard negatives and ensures that different negatives are explored across epochs, fostering better generalization. A key advantage of this approach lies in the gradual annealing of the temperature parameter $\tau'$ during fine-tuning. As training progresses, we decrease $\tau'$, thereby sharpening the softmax distribution and progressively increasing the difficulty of the sampled negatives. This forms a curriculum learning strategy, where the model is initially exposed to easier negatives and progressively harder ones as it becomes more adept at distinguishing semantically similar pairs.

To fine-tune the retriever, we employ a contrastive learning objective based on the InfoNCE loss (Oord et al., 2018). The objective seeks to maximize the similarity between the text $h_i$ and its positive counterpart $h_i^+$, while minimizing the similarity between the text $h_i$ and both hard negatives $H_B$ and other positives from other in-batch examples $H$. In our formulation, each positive has $N * (M + 1) - 1$ negatives in the contrastive loss. Specifically, the loss is formalized as:

$$\mathcal{L}_{CL}(\mathbf{h}_i, \mathbf{h}_{i+}) = -\log\left(\frac{\exp(\mathbf{h}_i \cdot \mathbf{h}_{i+}/\tau)}{\sum_{\mathbf{h}_k \in (\mathbf{H_B} \cup \mathbf{H})} \exp(\mathbf{h}_i \cdot \mathbf{h}_k/\tau)}\right) \quad (1)$$

Here, $\tau(= 0.07)$ represents the temperature parameter that controls the sharpness of the softmax distribution, and the dot product $\mathbf{h}_i \cdot \mathbf{h}_k$ represents the cosine similarity between the text query and the code snippet in the joint embedding space. During inference, the cosine similarity between the text query and the code snippet is used as the relevance score to get a ranked ordering.

### 3.2 CODERANKLLM RERANKER

Recently, listwise reranking approaches (Pradeep et al., 2023; Reddy et al., 2024) have gained popularity for their ability to score multiple passages simultaneously, as opposed to pointwise Zhuang et al. (2023a;c) or pairwise Qin et al. (2023) reranking, where scoring is performed in isolation. Xian et al. (2023) demonstrate that listwise reranking benefits from contextually comparing multiple passages at once, which helps calibrate relevance scoring better. Furthermore, Sun et al. (2023) show that instruction-tuned LLMs can outperform traditional supervised cross-encoders (Nogueira et al., 2020; Zhuang et al., 2023b) in zero-shot reranking settings. Due to input size limits, listwise reranking with LLMs usually adopts a sliding window strategy (Sun et al., 2023) with a window size of $M$ candidates and a stride $s$. For each window, passages are denoted by unique identifiers $y_i$; the LLM reranker generates as output a sequence of identifiers in decreasing order of their relevance.

However, CORNSTACK cannot be directly used for finetuning listwise rerankers, as they require ranked ordering data as supervision for training. Following recent work by Pradeep et al. (2023),

---

[6]We use $\gamma = 0.95$ with $\tau'$ linearly decayed from 0.05 to 0.001

we leverage larger LLMs as teacher models to train our listwise code reranker. The relevance supervision is provided in the form of an ordered sequence $y = y_1 > y_2 > ... > y_m$, where $y_i$ is the identifier of a document that has been judged more relevant to the query $q$ than $y_j$, for every $m \geq j > i$. Here, $\{y_i\}_{i=1}^{m}$ are taken from the positive and hard negative code snippets from CORN-STACK for each text query. The reranker is trained using the top $M$ negatives from the offline stage (in §2.3), since it is relies on the ranking ordering supervision for these negatives provided by the teacher model, which is not feasible to obtain in an online fashion. We then train the reranker with a language modeling objective, minimizing the error in predicting the true next token in the generation sequence:

$$\mathcal{L}_{LM} = -\sum_{i=1}^{|y|} \log(P_\theta(y_i|x, y_{<i})) \tag{2}$$

$P_\theta(y_i|x, y_{<i})$ is the conditional probability of predicting the target $y_i$ given the instruction prompt $x$ and the preceding tokens $y_{<i}$.

## 4    EXPERIMENTS

CORNSTACK is a high-quality curated code dataset containing <query, positive, negatives> tuples across six programming languages: Python, Java, Javascript, Ruby, Go, and PHP. This work aims to investigate two research questions: **RQ1:** Can CORNSTACK be leveraged to train highly performant code retrievers and rerankers?; **RQ2:** Can such a code retriever + reranker framework be used to assist in real-world software development? To address *RQ1*, we first demonstrate in §4.1 the superior performance of our code retriever on a variety of code retrieval tasks. Subsequently, in §4.2, we show the improved ranking accuracy achieved by leveraging our listwise code reranker over the retrieved results. For *RQ2*, §4.3 shows better function localization based on GitHub issues from using our code retriever + reranker framework.

### 4.1    CODE RETRIEVAL

#### 4.1.1    SETUP

**Training**   We finetune our code retriever using the 21 million contrastive examples in CORN-STACK. The encoder is initialized with Arctic-Embed-M (Merrick et al., 2024), a text encoder supporting an extended context length of 8,192 tokens and pretrained on large-scale web query-document pairs, along with public text retrieval datasets (Yang et al., 2018; Kwiatkowski et al., 2019; Thorne et al., 2018). We finetune for three epochs using four GH200 GPUs, with a batch size of 128 and 15 hard negatives per example. Our data filtering, negative mining, and model finetuning are implemented using the contrastors package (Nussbaum et al., 2024).

**Evaluation Datasets**   To demonstrate the effectiveness of CORNSTACK, we evaluate our fine-tuned retriever on a variety of code retrieval tasks under zero-shot settings. First, we consider Code-SearchNet (CSN) (Husain et al., 2019) and AdvTest (Lu et al., 2021) as benchmarks for function-level text-to-code retrieval, a semantic search task where natural language queries are used to retrieve relevant code snippets. Additionally, to evaluate performance across diverse code retrieval tasks, we consider the CoIR benchmark (Li et al., 2024), which includes code-to-text, code-to-code, and hybrid code retrieval tasks (retrieving a hybrid of code and textual documents given a hybrid query), in addition to text-to-code retrieval.

**Baselines**   We compare our finetuned code retriever against state-of-the-art open-source and proprietary text and code embedding models of various parameter sizes. For open-source text embedding models, we include E5-Base (Wang et al., 2022) and E5-Mistral (Wang et al., 2023a), the two most performant text embedding models from the CoIR benchmark, as well as Arctic-Embed-M (Merrick et al., 2024), the base text encoder that we finetune on CORNSTACK. For open-source code embedding models, we consider the Small, Base, and Large variants of CodeSage (Zhang et al., 2024), along with CodeT5+ (Wang et al., 2023b) and Jina-Code-v2 (Günther et al., 2023), which are the current state-of-the-art code embedding models on text-to-code retrieval benchmarks. CodeSage is trained on an older version of the Stack (Kocetkov et al.), while CodeT5+ and Jina-Code-v2 use

| Retriever | Param. | CodeSeachNet | | | | | | | AdvTest | COIR |
| | | Python | Java | JS | PhP | Go | Ruby | Avg. | Python | Avg. |
|---|---|---|---|---|---|---|---|---|---|---|
| Arctic-Embed-M | 137M | 53.8 | 49.5 | 46.3 | 41.1 | 71.9 | 57.9 | 53.4 | 34.1 | 43.0 |
| CodeSage-Small | 130M | 64.4 | 63.2 | 60.0 | 54.7 | 77.7 | 63.2 | 64.9 | 41.3 | 54.4 |
| CodeSage-Base | 356M | 68.0 | 68.0 | 67.0 | 58.2 | 83.2 | 68.0 | 68.7 | 49.1 | 57.5 |
| CodeSage-Large | 1.3B | 70.8 | 70.2 | 69.5 | 61.3 | 83.7 | 71.9 | 71.2 | 52.7 | 59.4 |
| Jina-Code-v2 | 161M | 64.4 | 66.4 | 61.8 | 55.9 | 84.4 | 70.4 | 67.2 | 37.1 | 58.4 |
| CodeT5+ | 110M | 71.7 | 71.8 | 69.2 | 67.8 | 90.7 | 74.4 | 74.2 | 40.8 | 45.9 |
| OpenAI-Ada-002 | Unknown | 68.0 | 71.5 | 67.5 | 60.6 | 85.6 | 74.2 | 71.3 | 38.1 | 45.6 |
| Voyage-Code-002 | Unknown | 66.8 | 64.8 | 63.4 | 52.0 | 88.9 | 75.0 | 68.5 | - | 56.3 |
| CODERANKEMBED | 137M | **78.4** | **76.9** | **71.4** | **68.8** | **92.7** | **79.3** | **77.9** | **59.5** | **60.1** |

Table 3: Ranking performance (%) for different retrievers on various code retrieval benchmarks. We report the official metrics for each dataset: MRR@1000 for CodeSearchNet and Advtest, and nDCG@10 for COIR. Detailed task-wise numbers for COIR are in Table 8 in Appendix.

| Approach | CSN | AdvTest |
|---|---|---|
| Consistency Filtering + Softmax Sampling of Hard Negatives + Curriculum Learning | **72.7** | **50.8** |
| Consistency Filtering + Softmax Sampling of Hard Negatives | 72.3 | 49.4 |
| Consistency Filtering + Top-K Selection of Hard Negatives | 71.4 | 48.6 |
| Consistency Filtering | 63.3 | 39.2 |
| No Filtering | 56.7 | 37.6 |

Table 4: Ablations showing benefit of each of our proposed techniques. *None* represents using unfiltered Stack v2 examples with only in-batch negatives and no additional hard negatives.

the GitHub Code dataset for pretraining. We also include proprietary embedding models OpenAI-Ada-002 and Voyage-Code-002 in our evaluation.

### 4.1.2 RESULTS

Table 3 presents the code retrieval performance results for CodeSearchNet, AdvTest, and CoIR. The task-specific CoIR results are presented in Table 8 in the Appendix. On CodeSearchNet, our approach significantly outperforms all open-source and proprietary text and code embedding models, establishing a new state-of-the-art for text-to-code retrieval. Notably, despite being evaluated in a zero-shot setting, our code retriever achieves better performance than CodeT5+, which uses CodeSearchNet for contrastive finetuning. Further, our 137M parameter encoder outperforms the 1.3B CodeSage-Large model, which is ten times larger. In addition, on CoIR, our code retriever, despite being smaller than the majority of the baselines, consistently performs well across all the tasks, leading to the highest average performance. This demonstrates the robustness of our contrastive training data, with the trained model exhibiting superior cross-task generalization despite being trained exclusively for only text-to-code retrieval.

### 4.1.3 ABLATION STUDIES

Here, we conduct ablation studies for the code retriever using 10% of the training data from CORN-STACK. Specifically, we measure the benefit of our proposed techniques, namely curriculum learning during training, the use of hard negatives with a softmax-based sampling strategy and consistency filtering aimed at eliminating noisy positives. Table 4 shows the results from the ablation experiments. We can see that removing consistency filtering of positives or the use of hard negatives separately leads to significant drop in performance on both CodeSearchNet (CSN) and AdvTest. We also see the benefit of curriculum learning, along with using softmax-based sampling of hard negatives instead of top-K selection.

| Reranker | FT Data | CodeSeachNet | | | | | | | AdvTest |
|----------|---------|--------|------|------|------|------|------|------|--------|
| | | Python | Java | JS | PhP | Go | Ruby | Avg. | Python |
| None (Retriever Only) | Code | 78.1 | 76.6 | 68.9 | 69.9 | 91.6 | 80.3 | 77.7 | 56.9 |
| Qwen-2.5-Code (zero-shot) | - | 71.6 | 70.7 | 64.0 | 63.1 | 84.0 | 71.6 | 70.2 | 58.7 |
| Qwen-2.5-Text (finetuned) | Text | 80.0 | 78.1 | 73.2 | 69.8 | 92.0 | 79.9 | 78.8 | 66.4 |
| CODERANKLLM | Code | **81.7** | **80.5** | **76.2** | **72.4** | **92.3** | **81.8** | **80.5** | **69.1** |

Table 5: Ranking performance (MRR@100 in %) for different models from reranking top-100 retrieval results on function-level text-to-code retrieval datasets. Our code reranker is finetuned from Qwen-2.5-Code with code listwise data, while Qwen-2.5-Text is finetuned using text listwise data.

## 4.2 CODE RERANKING

### 4.2.1 SETUP

**Training** To create the training data for listwise reranking, we pick 50k <query, positive, negatives> tuples from CORNSTACK by filtering for a higher similarity score and a better rank for the positive. Following Pradeep et al. (2023), a sampling strategy with varying window sizes (between 3 to 10) and random shuffling leads to 250k training instances (more details in §A.7 of Appendix). For ranking supervision, we use the Qwen-2.5-32B-Instruct LLM (Yang et al., 2024) to obtain the ranked ordering of each example. The Qwen-2.5-Coder-7B-Instruct model (Hui et al., 2024), which specializes in instruction-based code generation, is employed as the listwise reranker. We finetune this model for one epoch using four GH200 GPUs, with a batch size of 64 and a maximum input sequence length of 16,800.

**Baselines and Evaluation** We compare our reranking performance with the zero-shot Qwen-2.5-Coder-7B-Instruct model, which was used for fine-tuning. Since most text-based LLMs are trained on both text and code data, we include a listwise text reranker as a baseline. Specifically, we finetune the Qwen-2.5-7B-Instruct LLM using 40k listwise reranking instances labeled by GPT-4, as described in Pradeep et al. (2023), which were created using queries from the MS MARCO dataset (Nguyen et al., 2016). For evaluation, we employ the CodeSearchNet and AdvTest text-to-code retrieval benchmarks. However, we exclude the CoIR benchmark due to its significantly larger size (containing more than 100k queries). During inference, the top 100 results from our code retriever are passed to the reranker, with evaluation conducted using MRR@100. We use a window size of 10 and a step size of 5 for the listwise LLM rerankers.

### 4.2.2 RESULTS

Table 5 presents the performance of different reranking models on text-to-code retrieval datasets. Interestingly, the text reranker (Qwen-2.5-Text) demonstrates strong performance across multiple programming languages, despite being finetuned with listwise text reranking data. This performance is likely due to the presence of code examples in the LLM pretraining data, which enhances the model's understanding of code. Although the code LLM (Qwen-2.5-Text) performs worse in a zero-shot setting for listwise reranking, its performance improves significantly after finetuning with code-specific listwise data derived from CORNSTACK. These results suggest that listwise code rerankers can further enhance ranking performance beyond the initial retrieval step.

## 4.3 CODE RETRIEVAL+RERANKING FOR FUNCTION LOCALIZATION

Having previously evaluated our code retrieval and reranker models on academic benchmarks, we now demonstrate their utility in assisting software development in real-world settings. Specifically, we focus on the task of function localization, which involves accurately identifying the specific functions that need to be modified in response to a bug report or a GitHub issue.

| Model | Param. | File-level | | | Function-level | |
|---|---|---|---|---|---|---|
| | | Top-1 | Top-2 | Top-3 | Top-5 | Top-10 |
| E5-Base | 110M | 50.0 | 66.1 | 71.9 | 39.4 | 52.2 |
| Arctic-Embed-M | 137M | 45.7 | 63.2 | 68.2 | 40.5 | 49.6 |
| CodeSage-Small | 130M | 47.8 | 65.0 | 71.9 | 40.9 | 51.1 |
| CodeSage-Base | 356M | 48.5 | 66.4 | 74.1 | 39.8 | 48.9 |
| CodeSage-Large | 1.3B | 49.6 | 67.2 | 71.2 | 40.1 | 48.2 |
| Jina-Code-v2 | 161M | 42.0 | 62.4 | 70.1 | 43.1 | 55.1 |
| CodeT5+ | 110M | 47.8 | 62.4 | 68.2 | 39.8 | 45.3 |
| Agentless GPT-4o-mini | Unknown | 54.0 | 63.5 | 68.2 | 35.0 | 36.9 |
| Agentless GPT-4o | Unknown | 65.7 | 74.8 | 77.4 | 44.5 | 45.6 |
| CODERANKEMBED (Ours) | 137M | 51.8 | 68.6 | 76.6 | 50.0 | 59.1 |
| + CODERANKLLM (Ours) | 7B | **68.2** | **81.4** | **85.0** | **67.5** | **73.7** |

Table 6: File and function localization performance (%) on SWE-Bench-Lite.

### 4.3.1 SETUP

**Datasets**  For evaluation, we utilize SWE-Bench-Lite (Jimenez et al., 2024), a widely used evaluation suite for automated software engineering. SWE-Bench-Lite is a repository-level benchmark that focuses on resolving real-world issues sourced from GitHub, requiring a code patch that passes the associated test cases. Following Xia et al. (2024), we reformulate SWE-Bench-Lite for function localization evaluation by considering the functions to which code patches have been applied as the localized functions. Specifically, we retained 274 of 300 examples where patches modify existing functions or classes, with the excluded examples introducing code corresponding to new functions or import statements. The GitHub issue serves as the text query, while all functions within the files in the repository are considered as candidates for retrieval.

**Baselines and Metrics**  Our primary baseline is Agentless (Xia et al., 2024), an automated approach to solving software development problems that ranks among the top-performing open-source submissions on SWE-Bench-Lite. Agentless employs a two-phase process of localization followed by repair. In the localization phase, it uses a hierarchical approach to first localize the fault to specific files, then to relevant classes or functions, and finally to fine-grained edit locations. Given the considerable size of the codebase, a tree-like structure of the repository, illustrating the relative location of each file, along with the GitHub issue, is used to rank and identify the files that need edits. Subsequently, the content of these files is used to identify the functions within them that require modification. For a detailed description of Agentless, we refer the reader to Xia et al. (2024). We evaluate function localization using the output logs of Agentless obtained from the released official run. Furthermore, given the functions identified for modification, we map them to the files they belong to for file localization evaluation. Since Agentless selects up to three files that need edits and further localizes functions within them, we evaluate file localization at top 1–3 and function localization at top 5 and top 10. We also consider the retrieval baselines as in Section 4.1, except for the proprietary ones due to API costs.

### 4.3.2 RESULTS

Table 6 presents the function and file localization accuracy achieved on SWE-Bench-Lite. Results indicate that our code retriever significantly outperforms Agentless and other retrieval baselines on function localization. Additionally, we observe consistent improvements in both file and function localization when leveraging our code reranker on top of the retriever results. We hypothesize that the superior performance of GPT-4o on file localization, compared to the code retriever, may be due to these models having been exposed to the codebases during training, as SWE-Bench-Lite is constructed using popular open-source Python repositories. Therefore, GPT-4o can potentially identify the file to be edited without even utilizing the corresponding file content. We hypothesize that our retrieval-based approach could achieve further improvements on private repositories, which are typically not included in LLM pretraining data. We leave this investigation for future work.

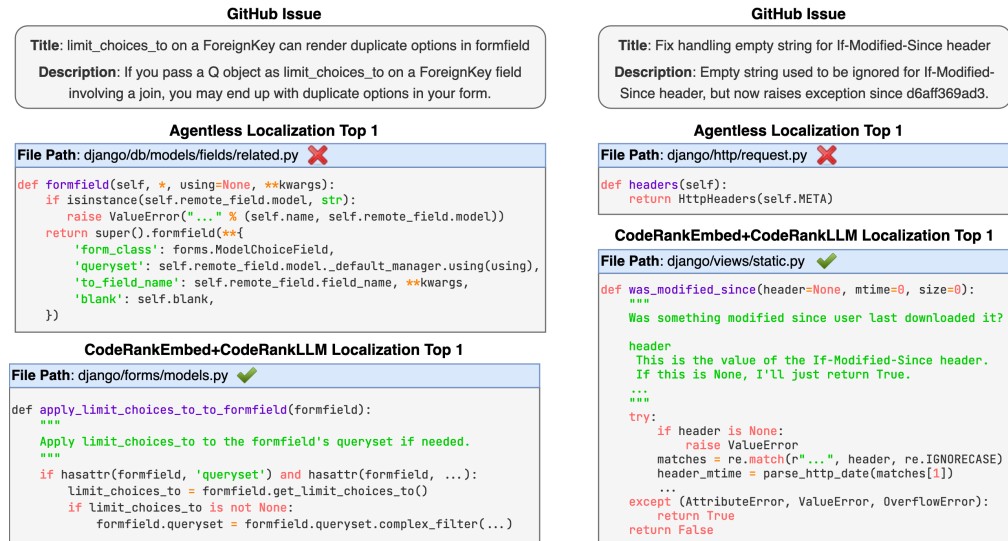

Figure 2: Examples from SWE-Bench-Lite where Agentless mislocalizes the file (and function), while our CODERANKEMBED+CODERANKLLM framework does function localization correctly.

Figure 2 illustrates examples from SWE-Bench-Lite where the Agentless system localizes incorrectly, likely influenced by the file naming conventions. In the left example, the system points to the *fields* folder, possibly because the issue mentions *formfield*. Similarly, in the right example, it selects the *request* file within the *http* folder, as the issue pertains to a problem with the header. In contrast, our framework accurately performs function-level localization in both cases. This is attributed to the retriever and reranker integrating information from both the docstring and the code snippet, allowing it to be correctly matched to the GitHub issue serving as the query.

## 5 CONCLUSION

This paper presents CORNSTACK, a large-scale, high-quality dataset of contrastive training instances for code retrieval and reranking. Fine-tuning embedding models on this contrastive data achieves state-of-the-art performance across various code retrieval tasks, outperforming code embedding models that are ten times larger. We also demonstrate that a listwise code reranker, fine-tuned using CORNSTACK, can further improve the code ranking accuracy. Moreover, using our code retriever and listwise code reranker together, we show significant improvements in function localization for GitHub issues, an important component of real-world software development.

## LIMITATIONS

Heuristic filtering in CoRNStack removed exact matches with evaluation datasets, but additional semantic filtering is needed to catch similar queries and code. To further validate the functional correctness of the code data, one could generate test cases using LLMs and/or human annotators. However, due to the dataset size, it would be computationally intensive to generate comprehensive test cases and set up the necessary environments for execution-based validation.

## ACKNOWLEDGEMENT

We would like to thank the BlenderNLP group and Nomic AI team for valuable feedback. This research is based upon work supported by DARPA ITM Program No. FA8650-23-C-7316 and DARPA SemaFor Program No. HR001120C0123. The views and conclusions contained herein are those of the authors and should not be interpreted as representing the official policies, either expressed or implied, of DARPA, or the U.S. Government. The U.S. Government is authorized to reproduce and distribute reprints for governmental purposes notwithstanding any copyright annotation therein.

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

# A APPENDIX

## A.1 DIVERSITY OF CODE TOPICS IN CORNSTACK

To provide more insight into the variety and complexity of code tasks in CoRNStack, we analyzed the distribution of code topics for 100k randomly sampled instances using Nomic Atlas[7], a popular unstructured text visualization tool. Nomic Atlas employs a cluster-based keyword identification algorithm and leverages a language model to generate topics. We find that the majority of examples fall into eight broad categories: object creation, data sorting, data management, API management, configuration, data validation, graphics, and math operations. The wordcloud in Figure 3 illustrates the diverse fine-grained topics within these categories.

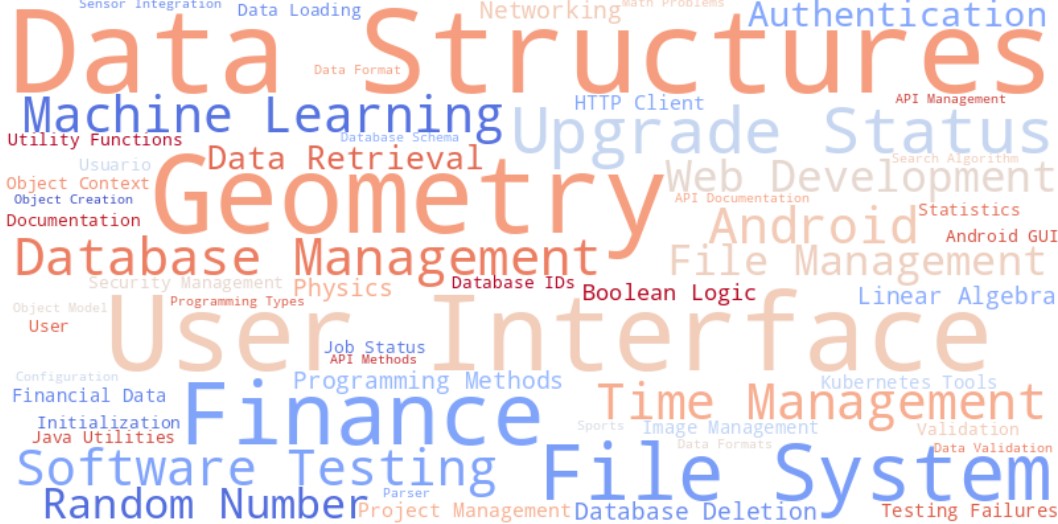

Figure 3: A wordcloud of popular code topics collected from 100k instances in CoRNStack

## A.2 ACCURACY OF (TEXT, CODE) PAIRINGS BY LANGUAGE

In Table 7, we provide the language-wise correctness numbers for the mean results from Table 2 in the main paper. We can see that CoRNStack has significantly higher correctness of (text, code) pairings across all languages.

| Dataset | Python | Java | JavaScript | PHP | Go | Ruby | Avg |
|---|---|---|---|---|---|---|---|
| Stack v2 | 54.8 | 50.3 | 53.6 | 56.4 | 63.2 | 39.2 | 52.9 |
| CSN | 53.9 | 56.3 | 50.5 | 60.7 | 65.9 | 47.6 | 55.8 |
| CosQA | 63.9 | - | - | - | - | - | 63.9 |
| CoRNStack (Ours) | **76.2** | **80.6** | **74.4** | **77.3** | **82.8** | **71.4** | **77.1** |

Table 7: Evaluation of <query, positive> pair correctness by language for different code corpora.

## A.3 FINE-TUNING VARIOUS ENCODERS ON CORNSTACK

We hypothesize that code retrievers can benefit from pretraining on supervised text ranking data, which is typically abundant. To validate this hypothesis, we performed the following experiment: We selected three ∼130M parameter text and code encoders, specifically Arctic-Embed-M Merrick et al. (2024) and Nomic-embed Nussbaum et al. (2024) as the text encoders due to their strong performance on text retrieval benchmarks like MTEB, and CodeSage-Small Zhang et al. (2024) as

---

[7]https://atlas.nomic.ai/data/corniclr25/cornstack-100k

| Task (→) | Text-to-Code | | | Code-to-Text | Code-to-Code | | | Hybrid Code | | | Avg |
|---|---|---|---|---|---|---|---|---|---|---|---|
| Retriever | Apps | CosQA | SQL | CSN | CSN -CCR | CodeTrans -Contest | -DL | StackOver Flow | CodeFeedBack -ST | -MT | |
| E5-base | 11.5 | 32.6 | 52.3 | 68.0 | 56.9 | 62.5 | 21.9 | 86.9 | 74.5 | 42.0 | 50.9 |
| Arctic-Embed-M | 5.4 | 27.6 | 18.9 | 37.4 | 62.2 | 68.9 | 28 | 86.9 | 67 | 28.0 | 43.0 |
| E5-Mistral | 21.3 | **51.3** | 66.0 | 54.3 | 65.3 | 82.6 | **33.2** | **91.5** | 72.7 | 33.7 | 55.2 |
| CodeSage-Small | 17.3 | 30.5 | 51.9 | 74.1 | 84.2 | 76.2 | 31.0 | 73.9 | 62.3 | 42.6 | 54.4 |
| CodeSage-Base | 27.6 | 29.4 | 59.4 | 76.9 | 86.9 | 78.9 | 31.9 | 76.2 | 63.0 | 44.6 | 57.5 |
| CodeSage-Large | **32.7** | 28.9 | 59.5 | 78.1 | **89.0** | 82.6 | 32.7 | 78.7 | 65.4 | **46.3** | 59.4 |
| Jina-Code-v2 | 16.4 | 42.2 | 46.4 | **84.0** | 82.7 | **83.6** | 26.8 | 89.3 | 68.6 | 44.4 | 58.4 |
| CodeT5+ | 3.3 | 23.1 | 41.1 | 78.0 | 83.6 | 52.3 | 31.6 | 59.9 | 53.2 | 32.8 | 45.9 |
| OpenAI-Ada-002 | 8.7 | 29.8 | 58.3 | 74.2 | 69.1 | 53.3 | 26.0 | 72.4 | 47.1 | 17.7 | 45.6 |
| Voyage-Code-002 | 26.5 | 29.8 | **69.3** | 81.8 | 73.5 | 72.8 | 27.3 | 77.7 | 65.4 | 28.7 | 56.3 |
| CODERANKEMBED | 21.1 | 36.3 | 58.8 | 83.7 | 86.9 | 78.8 | 32.8 | 82.3 | **75.7** | 45.2 | **60.1** |

Table 8: NDCG@10 for different retrievers on the Code Information Retrieval Benchmark (CoIR).

the code encoder due to its overall performance on code retrieval benchmarks. We evaluated these models on code retrieval tasks before and after finetuning on CoRNStack for one epoch. As shown in Table 9, we see that finetuning on CoRNStack significantly improves code retrieval performance for all models. Moreover, we observe that a stronger text embedding model (Arctic-Embed-M here) leads to better code retrieval performance after finetuning on CoRNStack, even outperforming the code-pretrained CodeSage-Small. These results highlight that CoRNStack, being large-scale and high-quality, can be leveraged to finetune text encoders into performant code retrievers. Therefore, we chose Arctic-Embed-M as it provides a strong foundation from supervised text ranking pretraining, which, when combined with fine-tuning on CoRNStack, leads to superior code retrieval.

| Base Model | Pretrain | # Params | CSN Avg. | AdvTest | CoIR Avg. |
|---|---|---|---|---|---|
| CodeSage-Small | Code | 130M | 64.9 → 73.9 (+9.0) | 41.3 → 54.2 (+12.9) | **54.4 → 60.0 (+5.6)** |
| Nomic-Embed | Text | 137M | 47.2 → 76.7 (+29.5) | 28.6 → 54.6 (+26.0) | 47.7 → 58.5 (+10.8) |
| Arctic-Embed-M | Text | 137M | **53.4 → 77.7 (+24.3)** | **34.1 → 57.8 (+23.7)** | 43.0 → 59.7 (+16.7) |

Table 9: Results (Before → After) from finetuning different encoders for 1 epoch on CoRNStack.

## A.4 EFFICACY OF FINE-TUNING ON CORNSTACK VS CODESEARCHNET

CoRNStack has a significantly higher query-positive correctness, while also being upto 10x larger than existing contrastive code datasets like CodeSearchNet (CSN) (see Table 2 in the main paper). To further highlight the impact of CoRNStack's scale and quality, we finetuned Arctic-Embed Merrick et al. (2024), a text embedding model, for one epoch separately on CoRNStack and CodeSearchNet. To specifically show the benefit of CoRNStack's quality, we also report results for fine-tuning on 2 million randomly sampled datapoints from CoRNStack, the same amount of data as CodeSearchNet. The results, shown in Table 10, clearly illustrate the improvement in code retrieval performance from finetuning on CoRNStack with both a 2M subset and the full 21M examples.

| Training Dataset | # Examples | CSN Avg | AdvTest |
|---|---|---|---|
| CodeSearchNet | 2M | 65.8 | 37.5 |
| CoRNStack Subset (Ours) | 2M | 71.4 | 48.6 |
| CoRNStack (Ours) | **21M** | **77.7** | **57.8** |

Table 10: Comparison of fine-tuning Arctic-Embed on CoRNStack vs CodeSearchNet.

## A.5 DETAILED EXPLANATION OF COMPARISON WITH CODET5+

In our paper, we compared our code retriever—an encoder-only model—to the publicly available 110M parameter CodeT5+ Embedding model[8] (denoted as CodeT5+ in our paper). This model is also encoder-only and is listed in the official CodeT5+ repository[9], but not discussed in the CodeT5+ paper Wang et al. (2023b). The repository also includes the 220M parameter CodeT5+ bimodal model[10], an encoder-decoder trained with text-code matching. CodeT5+ bimodal uses the 110M parameter CodeT5+ embedding model as its encoder and incorporates an additional decoder for reranking the top 32 candidates retrieved by the embedding model. Since CodeT5+ bimodal primarily serves as a reranker over the embedding model's results, we did not include this model in the comparison in our paper. Our results (in Table 3 of the main paper) closely align with the performance metrics of the CodeT5+ Embedding model reported in the CodeT5+ README. Additionally, the CodeT5+ authors note that the released CodeT5+ models are trained with multi-task data, and results differ from those in their paper, which are fine-tuned for each retrieval benchmark. Table 11 shows a detailed comparison of our code retriever with both the released multi-task CodeT5+ models and the single-task CodeT5+ models. We observe that our zero-shot code retriever outperforms both variants.

| Model | # Params | Model | Type | CSN | AdvTest |
|---|---|---|---|---|---|
| CodeT5+ Embedding | 110M | Retriever | Multi-Task Finetune | 74.2 | 40.8 |
| CodeT5+ Bimodal | 220M | Reranker | Multi-Task Finetune | 75.9 | 42.9 |
| CodeT5+ Bimodal | 220M | Reranker | Task-Specific Finetune | 77.1 | 43.3 |
| CodeT5+ Bimodal | 770M | Reranker | Task-Specific Finetune | 77.4 | 44.7 |
| CODERANKEMBED | 137M | Retriever | Zero-shot | **77.9** | **59.5** |

Table 11: Code Retriever Fine-Tuned on CoRNStack vs. Different CodeT5+ Models

## A.6 STUDENT VS TEACHER PERFORMANCE FOR LISTWISE CODE RERANKING

Here, we evaluate the teacher model for listwise code reranking, to see whether the student model reaches the teacher's performance after finetuning. Due to computational limitations in running the large teacher model on the entire CodeSearchNet (CSN) and AdvTest benchmarks, we evaluated on 1,000 random sampled queries for AdvTest and each language in CSN. Table 12 compares the performance of our finetuned Qwen 2.5 7B student model with the Qwen 2.5 32B teacher model. We observe that while the finetuned student slightly outperforms the teacher on CSN, it still shows a slight performance gap on the more challenging AdvTest benchmark.

| Reranker | CodeSeachNet | | | | | | | AdvTest |
|---|---|---|---|---|---|---|---|---|
| | Python | Java | JS | PhP | Go | Ruby | Avg. | Python |
| None (Retriever Only) | 76.3 | 77.4 | 71.1 | 70.9 | 91.4 | 80.2 | 77.9 | 58.0 |
| Qwen-2.5-7B (Student Model) | 70.9 | 71.8 | 66.0 | 64.7 | 83.6 | 72.4 | 71.5 | 60.6 |
| Qwen-2.5-32B (Teacher Model) | 79.8 | 80.4 | 76.8 | **73.3** | 92.0 | **82.6** | 80.8 | **73.6** |
| CODERANKLLM | **79.9** | **81.9** | **77.9** | 73.2 | **92.5** | 81.8 | **81.2** | 70.4 |

Table 12: Ranking performance (MRR@100 in %) on 1k randomly sampled queries (per language) from CodeSearchNet and AdvTest for teacher model vs student model before and after finetuning.

## A.7 DETAILS ON DATA AUGMENTATION FOR LISTWISE RERANKING

We follow the data augmentation methodology from Pradeep et al. (2023), incorporating techniques to create a more diverse and challenging training setup in order to obtain a more robust trained reranker. Specifically, the augmentation process consists of two key components:

---

[8]https://huggingface.co/Salesforce/codet5p-110m-embedding
[9]https://github.com/salesforce/CodeT5/tree/main/CodeT5+
[10]https://huggingface.co/Salesforce/codet5p-220m

- **Variable Window Sizes:** For each training instance, a random subset of candidate code snippets (ranging from 3 to 10 candidates) is sampled from the original ranked list to pass as input to the teacher model. This introduces variability in the input size and diversifies complexity of the reranking task, ensuring that the reranker model encounters a broader range of scenarios during training.

- **Random Shuffling:** To enhance the model's generalization ability across different document orders—beyond the default order provided by the retriever—random permutations are applied to the sampled windows. This technique has demonstrated effectiveness in traditional text reranking tasks (Pradeep et al., 2023), and we extend it to code reranking.

