# OpenReview forum: "CoRNStack: High-Quality Contrastive Data for Better Code Retrieval and Reranking"
_ICLR.cc/2025/Conference — ICLR 2025 Poster_

### Official Review · Reviewer_ZoFm · 2024-10-25

**Soundness:** 2
**Presentation:** 2
**Contribution:** 2
**Rating:** 5
**Confidence:** 4

**Summary:**

The article introduces CORNSTACK, a novel, high-quality dataset for contrastive code retrieval and reranking training. The primary motivation is to address the limitations in existing code embedding models, which struggle with complex, real-world software development tasks due to the noisy and inconsistent nature of current datasets. These models, crucial for bug localization and retrieval-augmented code generation, often fail to generalize beyond small, controlled tasks. The effectiveness of the dataset was demonstrated through the training of the retriever and reranker models.

**Strengths:**

1. The authors try to filter the high-quality code retrieval dataset from the stack dataset to solve the noise problem existing in the current code retrieval dataset.
2. The authors train both retriever and reranker models, and the final experimental results prove the effectiveness of the authors' method and achieve state-of-the-art results.
3. The authors evaluate the code retrieval dataset and also try to evaluate the code retrieval model for the error localization dataset.

**Weaknesses:**

1. This paper's contribution is mainly incremental. It improves data quality through filtering and negative sample mining methods on the Stack V2 dataset, which has been extensively explored in previous research work [1,2]. This paper only makes minor improvements based on these papers.
2. The paper lacks a more detailed comparison and analysis between the final obtained and original datasets beyond just quantity (such as case studies and other statistical conclusions).
3. In the writing of the paper, the construction of the dataset and the harmful sample mining methods (especially the curriculum learning technique, which is not model-independent) are coupled. In my opinion, these are two parts: one part is the construction of the dataset, and the other part is the harmful sample mining method designed by the author. I believe these two parts should be more clearly distinguished in the writing.
4. For the codet5+ model, the results reported in the paper seem to differ from those presented in the original text. Could the author add more detailed experimental settings?
[1]Zhang, Dejiao, et al. "Code representation learning at scale." arXiv preprint arXiv:2402.01935 (2024).
[2] Liang Wang, Nan Yang, Xiaolong Huang, Binxing Jiao, Linjun Yang, Daxin Jiang, Rangan Majumder, and Furu Wei. Text embeddings by weakly-supervised contrastive pre-training.

**Questions:**

Please refer to Weaknesses section.

---

> ### Author Response · Authors · 2024-11-19
> **Response to Reviewer ZoFm (1/3)**
>
> We thank the reviewer for their valuable feedback, and we address below their concerns and comments:
>
> ***Weakness 1: This paper's contribution is mainly incremental. It improves data quality through filtering and negative sample mining methods on the Stack V2 dataset, which has been extensively explored in previous research work [1,2]. This paper only makes minor improvements based on these papers.***
>
> **Response:**
> We would like to clarify that our contributions extend significantly beyond incremental improvements:
>
> **1. Release of largest public high-quality contrastive code data:** We introduce CoRNStack, which is ten times larger than CodeSearchNet—the largest existing public contrastive code training dataset. Also, we are the first to apply consistency filtering to enhance data quality for contrastive fine-tuning of code embedding models. This results in CoRNStack having significantly higher query-positive correctness compared to existing contrastive code datasets like CodeSearchNet and CosQA (see Table 2 in the paper). To further highlight the impact of CoRNStack’s scale and quality, we finetuned Arctic-Embed, a text embedding model, for one epoch separately on CoRNStack and CodeSearchNet. To specifically show the benefit of CoRNStack’s quality, we also report results for fine-tuning on 2 million randomly sampled datapoints from CoRNStack, the same amount of data as CodeSearchNet. The results, shown in the table below, clearly illustrate the improvement in code retrieval performance from finetuning on CoRNStack with both a 2M subset and the full 21M examples. We also added this comparison to Section A.5 of the Appendix.
>
> | Training Dataset | # Examples | CSN Avg | AdvTest |
> |-----------------|------------|---------|----------|
> | CodeSearchNet | 2M | 65.8 | 37.5 |
> | CoRNStack Subset (Ours) | 2M | 71.4 | 48.6 |
> | CoRNStack (Ours) | **21M** | **77.7** | **57.8** |
>
> Moreover, unlike state-of-the-art open-source code embedding models, such as CodeSage and Jina-Code-v2 , which only provide model weights without releasing their training data, CoRNStack is publicly available at the following [anonymized link](https://huggingface.co/corniclr25). This establishes CoRNStack as a standardized high-quality dataset for enabling future research in code retrieval and reranking.
>
> **2. First to finetune LLMs as listwise code rerankers:** While listwise reranking has gained popularity in text retrieval [1], its application in code ranking has been largely unexplored, due to the lack of high-quality contrastive bimodal (text, code) data for fine-tuning. In section 3.2 of our paper, we have demonstrated how CoRNStack, with the curated set of (text, code) pairs that involve positives and negatives, can be leveraged to finetune instruction-tuned code generation models as effective listwise code rerankers. Tables 6 and 8 in the paper show significant performance improvement from using our code reranker on top of results from our code retriever.
>
>
> **3. Demonstrate the benefit of improved code retrieval & reranking for real-world software development:** Our work specifically addresses the gap between existing code ranking models and their practical application in complex software development tasks like repository-level function localization. Prior work, such as CodeRAG-Bench [2], hasn’t seen much success from leveraging code embedding models on complex tasks such as bug localization on SWE-Bench-Lite. In this paper, we show that this gap is largely due to a lack of high-quality contrastive data. We demonstrate that combining our code retriever & reranker setup, which were finetuned on CoRNStack, shows significant improvements in repository-level bug localization over Agentless [3], a top-performing automated software development framework.
>
> [1] RankZephyr: Effective and Robust Zero-Shot Listwise Reranking is a Breeze!; Pradeep et al.
>
> [2]  “Coderag-bench: Can retrieval augment code generation?”; Wang et al., arXiv 2024
>
> [3] “Agentless: Demystifying llm-based software engineering agents”; Xia et al., arXiv 2024

---

> > ### Author Response · Authors · 2024-11-25
> > **Following Up on Review**
> >
> > Dear Reviewer ZoFm,
> >
> > Thank you for your detailed and constructive feedback. We appreciate you actively engaging us in the rebuttal period, and revising the scores in reflection of our comprehensive rebuttal.
> >
> > Please let us know if you have any additional questions or concerns, and we look forward to your final score/feedback.
> >
> > Thanks,
> > ICLR 2025 Conference Paper 12599 Authors

---

> > > ### Comment · Reviewer_ZoFm · 2024-11-25
> > >
> > > Thank you for your responses. I maintain my final score. In summary, I think this is a labor-intensive work. However, the technical innovations, such as changing top-k negative sampling to softmax negative sampling and incorporating curriculum learning, are limited to me.

---

> > > > ### Author Response · Authors · 2024-11-28
> > > >
> > > > We sincerely thank the reviewer for their engagement during the discussion period. We appreciate their recognition of our domain contributions, particularly the release of the largest, high-quality contrastive text-code dataset, which we show is a valuable resource for training state-of-the-art code retrievers and rerankers.
> > > >
> > > > Regarding our technical contributions, we believe that our proposed softmax-based negative sampling method and curriculum learning approach are both innovative. Our softmax-based sampling improves diversity of mined negatives, reducing overfitting to certain negatives as a result of high semantic duplication in public sources like GitHub. Our curriculum learning improves adaptation, especially for encoders not pre-trained on code. As demonstrated in our ablation experiments, these methods contribute meaningfully to performance improvements for our code retriever and have not been employed in prior work.

---

> ### Author Response · Authors · 2024-11-19
> **Response to Reviewer ZoFm (2/3)**
>
> ***Weakness 2: The paper lacks a more detailed comparison and analysis between the final obtained and original datasets beyond just quantity (such as case studies and other statistical conclusions).***
>
> **Response:**
>
> To assess the accuracy of the text-code pairings in CoRNStack, we conducted an automatic evaluation as described in lines 174–180 and shown in Table 2 of the original paper. In short, Qwen2.5-Coder-7B-Instruct, a code generation model, was prompted to judge whether the positive code snippet fully answers the corresponding query. This was performed on 10,000 randomly sampled pairs from CoRNStack, the original Stack v2, and other contrastive code training datasets like CosQA and CodeSearchNet (CSN), across three random seeds. The results indicate that CoRNStack has a significantly higher mean percentage of correct pairings. We further provide below the language-wise correctness for the mean numbers reported in the paper and add this table to section A.3 of the revised version.
>
> | Dataset            | Python | Java  | Javascript | PHP   | Go    | Ruby  | Avg  |
> |--------------------|--------|-------|------------|-------|-------|-------|------|
> | Stack v2          | 54.8   | 50.3  | 53.6       | 56.4  | 63.2  | 39.2  | 52.9 |
> | CSN               | 53.9   | 56.3  | 50.5       | 60.7  | 65.9  | 47.6  | 55.8 |
> | CosQA             | 63.9   | -     | -          | -     | -     | -     | 63.9 |
> | **Ours (CoRNStack)**  | **76.2**   | **80.6**  | **74.4**       | **77.3**  | **82.8**  | **71.4**  | **77.1** |
>
> **Weakness 3: In the writing of the paper, the construction of the dataset and the harmful sample mining methods (especially the curriculum learning technique, which is not model-independent) are coupled. In my opinion, these are two parts: one part is the construction of the dataset, and the other part is the harmful sample mining method designed by the author. I believe these two parts should be more clearly distinguished in the writing.**
>
> **Response:**
>
> We sincerely thank the reviewer for this valuable suggestion. We agree that the curriculum learning component, which is related to the model training process, can be separated from the dataset construction description. We have updated the paper accordingly in the revised version. Specifically, we have moved the curriculum learning details to Section 3.1 (changes highlighted in blue). We hope that this revision adequately addresses the reviewer’s concern.

---

> > ### Author Response · Authors · 2024-11-19
> > **Response to Reviewer ZoFm (3/3)**
> >
> > **Weakness 4: For the codet5+ model, the results reported in the paper seem to differ from those presented in the original text. Could the author add more detailed experimental settings?**
> >
> > **Response:**
> >
> > In our paper, we compared our code retriever—an encoder-only model—to the publicly available 110M parameter [CodeT5+ Embedding](https://huggingface.co/Salesforce/codet5p-110m-embedding) model (denoted as CodeT5+ in our paper). This model is also encoder-only and is listed in the official [CodeT5+ repository](https://github.com/salesforce/CodeT5/tree/main/CodeT5%2B), but not discussed in the CodeT5+ paper [1]. The repository also includes the 220M parameter CodeT5+ bimodal model, an encoder-decoder trained with text-code matching. CodeT5+ bimodal uses the 110M parameter CodeT5+ embedding model as its encoder and incorporates an additional decoder for reranking the top 32 candidates retrieved by the embedding model. Since CodeT5+ bimodal primarily serves as a reranker over the embedding model’s results, we did not include this model in the comparison in our paper. Our results (in Table 3 of the paper) closely align with the performance metrics of the CodeT5+ Embedding model reported in the CodeT5+ [README](https://github.com/salesforce/CodeT5/blob/main/CodeT5%2B/README.md#evaluation-results). Additionally, the CodeT5+ authors note that the released CodeT5+ models are trained with multi-task data, and results differ from those in their paper, which are fine-tuned for each retrieval benchmark.
> >
> > In response to the reviewer’s question, we have added a detailed comparison in Section A.6 of the Appendix in the revised paper, including both the released multi-task CodeT5+ models and the single-task CodeT5+ 220M results, as shown in the table below. We observe that our zero-shot code retriever outperforms both variants.
> >
> > | Model | # Params | Model | Type | CSN | AdvTest |
> > |-------|----------|-------|------|-----|---------|
> > | CodeT5+ Embedding | 110M | Retriever | Multi-Task Finetune | 74.2 | 40.8 |
> > | CodeT5+ Bimodal | 220M | Reranker | Multi-Task Finetune | 75.9 | 42.9 |
> > | CodeT5+ Bimodal | 220M | Reranker | Task-Specific Finetune | 77.1 | 43.3 |
> > | CodeT5+ Bimodal | 770M | Reranker | Task-Specific Finetune | 77.4 | 44.7 |
> > | Ours | 137M | Retriever | Zero-shot | **77.9** | **59.5** |
> >
> >
> > [1] CodeT5+: Open Code Large Language Models for Code Understanding and Generation; Wang et al EMNLP 2023.

---

> > > ### Comment · Reviewer_ZoFm · 2024-11-20
> > >
> > > I appreciate the author's response. In the current version of the paper, the author has addressed my concerns about writing (weakness 3) and the comparison with the baseline (weakness 4). However, I think the author has not directly responded the weakness 1 about the lack of contribution. I can divide the paper's contributions into two parts:
> > >
> > > + Domain Contribution: The author has ultimately released a high-quality, large-scale code retrieval dataset and designed various methods to validate its effectiveness (training retrievers and rankers, evaluating in error localization problems closer to real-world scenarios).
> > > + Technical Contribution: I still believe the author's contributions in methodology are incremental, as they have been extensively explored in previous research [1,2]. Moreover, since the author has not conducted detailed ablation experiments on the proposed techniques about consistency filtering, softmax-based negative sampling, and curriculum learning, it is difficult for me to assess whether the dataset's quality is due to the methods proposed in the paper or the techniques from prior research.
> > >
> > > Accoding to the author's effective resolution of my concerns about writing and baseline comparisons, I will appropriately raise my score.

---

> > > > ### Author Response · Authors · 2024-11-23
> > > > **Response to Reviewer ZoFm**
> > > >
> > > > We thank the reviewer for raising their score. We address the remaining concerns below:
> > > >
> > > > ### **Ablation Experiments:** ###
> > > >
> > > > We have conducted additional ablation experiments focusing on our proposed techniques: **consistency filtering**, **softmax-based sampling of hard negatives**, and **curriculum learning**. These experiments augment the initial ablation studies (conducted with 10% of the training data which is 2.1 million training examples) presented in Section 4.1.3 (Table 5) of the original paper.
> > > >
> > > > The updated ablation table below highlights the individual impact of each technique. Specifically, we observe that both Curriculum Learning and Softmax-based Negative Sampling contribute to improvement in model performance. For clarity: the "None" row represents using unfiltered StackV2 examples with only in-batch negatives and no additional hard negatives.
> > > >
> > > > | Approach | CSN | AdvTest |
> > > > |----------|-----|---------|
> > > > | **Consistency Filtering + Softmax Sampling of Hard Negatives + Curriculum Learning** | **72.7** | **50.8** |
> > > > | Consistency Filtering + Softmax Sampling of Hard Negatives | 72.3 | 49.4 |
> > > > | Consistency Filtering + Top-K Selection of Hard Negatives | 71.4 | 48.6 |
> > > > | Consistency Filtering | 63.3 | 39.2 |
> > > > | None | 56.7 | 37.6 |
> > > >
> > > >
> > > >
> > > > We appreciate the reviewer’s feedback, which has helped strengthen the empirical validation of our work with these new ablation studies included in Section 4.1.3 and Table 5 accordingly in the revised version of the paper.
> > > >
> > > >
> > > > ### **Discussion on Technical Contributions:** ###
> > > >
> > > > While [1] uses only in-batch negatives with weighting (which is limited by the hardness of negatives available within a single batch), other approaches from text embedding literature introduce additional hard negatives with Top-K selection [2,3]. On the other hand, our work introduces a **softmax-based negative sampling strategy combined with curriculum learning**, specifically designed to address the unique challenges of contrastive learning for code embeddings. We highlight the benefits of these techniques below:
> > > >
> > > > 1. **Improved Exploration of the Negative Space with Softmax-based Sampling:** Code data inherently contains multiple semantically equivalent snippets, as a single functionality can have numerous valid implementations. This is particularly prevalent when sourcing code from open platforms like GitHub, where different developers may produce similar code. Traditional top-K selection of hard negatives can reduce diversity by over-representing only a few semantically similar examples, which risks overfitting during contrastive training. On the other hand, uniformly sampling from a larger K to improve diversity increases the presence of weak negatives. Our softmax-based negative sampling strategy thereby strikes a balance by improving the diversity of negatives without compromising their hardness.
> > > > 2. **Better Adaptation with Curriculum Learning:** Our curriculum learning strategy starts with easier negatives and gradually progresses to harder negatives as training advances. This approach is especially beneficial when fine-tuning embedding models pre-trained on text data but not previously exposed to code, facilitating smoother adaptation and better convergence.
> > > >
> > > >
> > > > Therefore, we believe our method not only addresses the challenge of handling semantically equivalent implementations—a fundamental issue in code data—but also introduces curriculum learning as a principled way to enhance generalization. Our experiments demonstrate that these contributions collectively improve the training of code embedding models, leading to state-of-the-art performance on various code retrieval benchmarks.
> > > >
> > > > We hope this clarifies the significance of our technical contributions and addresses your concerns regarding their novelty.
> > > >
> > > > [1] Zhang, Dejiao, et al. "Code representation learning at scale." ICLR 2024.
> > > >
> > > > [2] Liang Wang, Nan Yang, Xiaolong Huang, Binxing Jiao, Linjun Yang, Daxin Jiang, Rangan Majumder, and Furu Wei. Text embeddings by weakly-supervised contrastive pre-training.
> > > >
> > > > [3] Gabriel de Souza P. Moreira and Radek Osmulski and Mengyao Xu and Ronay Ak and Benedikt Schifferer and Even Oldridge. NV-Retriever: Improving text embedding models with effective hard-negative mining

---

### Official Review · Reviewer_bBWX · 2024-10-28

**Soundness:** 3
**Presentation:** 3
**Contribution:** 3
**Rating:** 6
**Confidence:** 3

**Summary:**

The paper constructs a dataset containing high-quality positive and negative samples for contrastive learning, thereby achieving better retrieval and reranking performance. They ensure a strong semantic correlation between the positive samples and queries in the dataset through filtering, and they obtain high-quality and diverse negative samples through hard negative mining. The model trained on this dataset achieves generally excellent retrieval performance across different benchmarks and retrieval tasks. Additionally, they further extend this dataset to build an ordered dataset for training the reranker, which also yields good results. Finally, they combine the retriever and reranker to achieve effective function localization for GitHub issues.

**Strengths:**

1. The paper is easy to follow

2. Motivation is clear

3. The model achieves effective results, showing significant improvements in various scenarios, including retrieval, reranking, and function localization.

**Weaknesses:**

There are some minor flaws in the experimental section that could be further improved:

1.The motivation for selecting Arctic-Embed-M as the initial encoder for training, instead of other models, is not clearly explained.

2.The setup in section 4.2.1 lacks clarity. For example, the statement “a sampling strategy with varying window sizes (between 3 to 10) and random shuffling leads to 250k training instances” could be elaborated more for better clarity.

3.In training the reranker, the authors used the QWen 32B model as a teacher to annotate data. It might be beneficial to include the teacher model’s performance in Table 6, allowing us to see if the model approaches the teacher’s performance on this task.

4.In section 2.3, the authors mentioned using a softmax-based sampling strategy instead of directly adopting top-k sampling for better diversity of negative samples. Including an ablation study for this strategy in the experimental section could more clearly validate the effectiveness of your approach.

**Questions:**

Refer to weakness 2. What's your motivation to select Arctic-Embed-M as your initial encoder rather than other models?

---

> ### Author Response · Authors · 2024-11-19
> **Response to Reviewer bBWX (1/2)**
>
> We appreciate the reviewer's positive feedback on our paper's clarity, motivation, and significant results across various scenarios, and we address here the concerns in the experimental section that they have suggested to further strengthen our work:
>
> ***Weakness 1: The motivation for selecting Arctic-Embed-M as the initial encoder for training, instead of other models, is not clearly explained.***
>
> **Response:**
>
> Our motivation for using Arctic-Embed-M as the initial encoder is based on the hypothesis that code retrievers can benefit, especially for text-code retrieval tasks, from pretraining on supervised text ranking data, which is typically abundant. To validate this hypothesis, we performed the following experiment:
>
> We selected three ~130M parameter text and code encoders, specifically Arctic-Embed-M and Nomic-Embed as the text encoders due to their strong performance on text retrieval benchmarks like MTEB, and CodeSage-Small as the code encoder due to its overall performance on code retrieval benchmarks. We evaluated these models on code retrieval tasks before and after finetuning on CoRNStack for one epoch. The results are presented in the table below and also in Section A.4 of the revised paper.
>
>
>
> | Base Model        | Pretrain Data | # Params | CSN Avg (Before -> After)    | AdvTest (Before -> After)    | ColR Avg (Before -> After)    |
> |-------------------|---------------|----------|------------------------------|------------------------------|-------------------------------|
> | CodeSage-Small    | Code          | 130M     | 64.9 -> 73.9 (+9.0)          | 41.3 -> 54.2 (+12.9)         | 54.4 -> **60.0** (+5.6)       |
> | Arctic-Embed-M    | Text          | 137M     | 53.4 -> **77.7** (+24.3)     | 34.1 -> **57.8** (+23.7)     | 43.0 -> 59.7 (+16.7)      |
> | Nomic-Embed       | Text          | 137M     | 47.2 -> 76.7 (+29.5)     | 28.6 -> 54.6 (+26)       | 47.7 -> 58.5 (+10.8)      |
>
> Firstly, we see that finetuning on CoRNStack significantly improves code retrieval performance for all models. Moreover, we observe that a stronger text embedding model (Artic-Embed-M in this case) leads to better code retrieval performance after finetuning on CoRNStack, even outperforming the code-pretrained CodeSage-Small in most cases. These results highlight that CoRNStack, being large-scale and high-quality, can be leveraged to finetune text encoders into performant code retrievers. Therefore, we selected Arctic-Embed-M as it provides a strong foundation from supervised text ranking pretraining, which, when combined with fine-tuning on CoRNStack, leads to superior code retrieval performance.
>
> ***Weakness 2: The setup in section 4.2.1 lacks clarity. For example, the statement “a sampling strategy with varying window sizes (between 3 to 10) and random shuffling leads to 250k training instances” could be elaborated more for better clarity.***
>
> **Response:**
>
> To clarify, we follow the data augmentation methodology established in prior work [1], incorporating techniques to create a more diverse and challenging training setup in order to obtain a more robust trained reranker. Specifically, the augmentation process consists of two key components:
>
> 1. Variable Window Sizes: For each training instance, a random subset of candidate code snippets (ranging from 3 to 10 candidates) is sampled from the original ranked list to pass as input to the teacher model. This introduces variability in the input size and diversifies complexity of the reranking task, ensuring that the reranker model encounters a broader range of scenarios during training.
> 2. Random Shuffling: To enhance the model's generalization ability across different document orders—beyond the default order provided by the retriever— random permutations are applied to the sampled windows. This technique has demonstrated effectiveness in traditional text reranking tasks [1], and we extend it to code reranking.
>
> We add these details to Section A.8 in the Appendix of the revised paper.
>
> [1] RankZephyr: Effective and Robust Zero-Shot Listwise Reranking is a Breeze!; Pradeep et al.

---

> > ### Author Response · Authors · 2024-11-19
> > **Response to Reviewer bBWX (2/2)**
> >
> > ***Weakness 3: In training the reranker, the authors used the QWen 32B model as a teacher to annotate data. It might be beneficial to include the teacher model’s performance in Table 6, allowing us to see if the model approaches the teacher’s performance on this task.***
> >
> > **Response:**
> >
> > We thank the reviewer for the valuable suggestion. In response, we have added Section A.7 in the Appendix, detailing the performance of the Qwen 2.5 32B teacher model on code reranking tasks. Due to computational limitations in running the large teacher model on the entire CodeSearchNet (CSN) and AdvTest benchmarks, we evaluated on random samples of 1,000 queries from AdvTest and each language in CSN. The results, presented in the table below, compare the performance of our fine-tuned Qwen 2.5 7B student model with the teacher model. We observe that while the student model slightly outperforms the teacher on CSN, it still shows a slight performance gap on the more challenging AdvTest benchmark.
> >
> > | Model Name                   | CSN Python | CSN Java | CSN Javascript | CSN PHP | CSN Go | CSN Ruby | CSN Avg | AdvTest |
> > |------------------------------|------------|----------|----------------|---------|--------|----------|---------|---------|
> > | Retriever                    | 76.3       | 77.4     | 71.1           | 70.9    | 91.4   | 80.2     | 77.9    | 58.0    |
> > | Qwen 2.5 7B (Base model)     | 70.9       | 71.8     | 66.0           | 64.7    | 83.6   | 72.4     | 71.5    | 60.6    |
> > | Qwen 2.5 32B (Teacher model) | 79.8       | 80.4     | 76.8           | **73.3**    | 92.0   | **82.6**     | 80.8    | **73.6**    |
> > | Finetuned Qwen 2.5 7B        | **79.9**       | **81.9**     | **77.9**           | 73.2    | **92.5**   | 81.8     | **81.2**    | 70.4    |
> >
> >
> > ***Weakness 4: In section 2.3, the authors mentioned using a softmax-based sampling strategy instead of directly adopting top-k sampling for better diversity of negative samples. Including an ablation study for this strategy in the experimental section could more clearly validate the effectiveness of your approach.***
> >
> > **Response:**
> >
> > Given the compute constraints, we have performed an ablation experiment with 500k training datapoints from using the top-k sampling vs using the curriculum-based softmax sampling. The table below reports numbers on CodeSearchNet (CSN) and AdvTest. We can see that the softmax-based sampling does provide better performance. We will add results from an updated run with more data in the final version of the paper.
> >
> >
> >
> > | Mining Strategy | CSN Avg | AdvTest |
> > |----------------|---------|---------|
> > | Top-k          | 67.5    | 44.9    |
> > | Softmax        | **69.6**    | **45.7**    |

---

> > > ### Author Response · Authors · 2024-11-23
> > > **Response to Reviewer bBWX**
> > >
> > > ### **Ablation Experiments for Weakness 4:** ###
> > >
> > > As an update to our response to weakness 4, we have conducted additional ablation experiments focusing on our proposed techniques: **consistency filtering**, **softmax-based sampling of hard negatives**, and **curriculum learning**. These experiments augment the initial ablation studies (conducted with 10% of the training data which is 2.1 million training examples) presented in Section 4.1.3 (Table 5) of the original paper.
> > >
> > >
> > > The updated ablation table below highlights the individual impact of each technique. Specifically, we observe that both Curriculum Learning and Softmax-based Negative Sampling contribute to improvement in model performance. For clarity: the "None" row represents using unfiltered StackV2 examples with only in-batch negatives and no additional hard negatives.
> > >
> > > | Approach | CSN | AdvTest |
> > > |----------|-----|---------|
> > > | **Consistency Filtering + Softmax Sampling of Hard Negatives + Curriculum Learning** | **72.7** | **50.8** |
> > > | Consistency Filtering + Softmax Sampling of Hard Negatives | 72.3 | 49.4 |
> > > | Consistency Filtering + Top-K Selection of Hard Negatives | 71.4 | 48.6 |
> > > | Consistency Filtering | 63.3 | 39.2 |
> > > | None | 56.7 | 37.6 |
> > >
> > >
> > > We appreciate the reviewer’s feedback, which has helped strengthen the empirical validation of our work with these new ablation studies included in Section 4.1.3 and Table 5 accordingly in the revised version of the paper.

---

> ### Author Response · Authors · 2024-11-25
> **Following Up on Review**
>
> Hello Reviewer bBWX,
>
> We would like to thank you for the thoughtful and constructive feedback on our submission.
>
> As the rebuttal period comes to a close, we are following up to inquire if our responses have addressed the concerns and would appreciate **you considering raising your final rating** or providing further feedback/concerns. We are looking forward to your response!
>
> Thanks,
> ICLR 2025 Conference Paper 12599 Authors

---

> > ### Author Response · Authors · 2024-11-28
> > **Following up again**
> >
> > We wanted to follow up again with the reviewer to inquire if our responses have addressed the concerns and would appreciate it if they consider raising their score or providing further feedback/concerns. We are looking forward to the reviewer’s response!
> >
> > Thanks,
> >
> > ICLR 2025 Conference Paper 12599 Authors

---

> > > ### Author Response · Authors · 2024-12-02
> > > **Follow-up Request**
> > >
> > > Given that today is the last date of the discussion period, we are following up again with the reviewer to inquire if our responses have addressed the concerns and would appreciate it if they consider raising their score or providing further feedback/concerns. We are looking forward to the reviewer’s response!
> > >
> > > Thanks,
> > >
> > > ICLR 2025 Conference Paper 12599 Authors

---

> > > > ### Comment · Reviewer_bBWX · 2024-12-03
> > > >
> > > > Thank you very much for the effort you put into the response, which addressed most of my concerns. I will maintain my score and stay positive toward the acceptance of this paper.

---

### Official Review · Reviewer_MeoD · 2024-10-30

**Soundness:** 3
**Presentation:** 3
**Contribution:** 2
**Rating:** 6
**Confidence:** 3

**Summary:**

This paper presents a dataset built upon The Stack V2 dataset, proposing a high-quality contrastive training dataset called CoRNStack, which consists of (text, code) pairs. This dataset uses dual consistency filtering to eliminate noisy positives and is further enriched with mined hard negatives. The effectiveness of the proposed dataset is demonstrated through downstream code tasks, including code retrieval and re-ranking tasks.

**Strengths:**

A new dataset has been established on existing code datasets by filtering and selecting positive and negative samples.

**Weaknesses:**

The process of handling negative samples in Figure 1 needs to be explained in more detail and more directly.

**Questions:**

1. The quality of the new dataset needs to be explained, especially how accurate the pairings are between text queries and code.
2. It should also be clarified whether the code data in the dataset has been checked manually or tested with examples, as this is important for the quality of the code retrieval tasks.
3. Additionally, it would be helpful to include information about the variety of code tasks in the new dataset, particularly how simple and complex scenarios are distributed.

---

> ### Author Response · Authors · 2024-11-19
> **Response to Reviewer MeoD (1/2)**
>
> We thank the reviewer for the valuable feedback and comments. We address the reviewer’s concerns and questions below:
>
>  ***Weakness 1: The process of handling negative samples in Figure 1 needs to be explained in more detail and more directly.***
>
> **Response:**
> Our hard negative mining strategy is divided into two stages:
> 1. An offline stage that is coupled with consistency filtering and computes a corpus-level similarity score matrix S between all text and code instances, which is used to filter false negatives.
> 2. An online stage that uses a softmax-based sampling strategy with a curriculum based on the pre-computed matrix S to progressively select diverse, challenging negatives during the contrastive finetuning.
>
>
> The retriever uses the negatives from the online stage to incorporate a curriculum learning strategy while training. The reranker, on the other hand, is trained using top-M negatives based on S from the offline stage, since it is relies on the ranking ordering supervision for these negatives provided by the teacher model, which is not feasible to obtain in an online fashion. We have made the distinction between the online and offline stages more clear in the revised version of the paper. The offline and online stages are described in more detail in lines 203-207 of Section 2.3, 224-235 of Section 3.1, and 267-269 of Section 3.2.
>
> ***Question 1: The quality of the new dataset needs to be explained, especially how accurate the pairings are between text queries and code.***
>
> **Response:**
>
> To assess the accuracy of the text-code pairings in CoRNStack, we conducted an automatic evaluation as described in lines 174–180 and shown in Table 2 of the original paper. In short, Qwen2.5-Coder-7B-Instruct, a code generation model, was prompted to judge whether the positive code snippet fully answers the corresponding query. This was performed on 10,000 randomly sampled pairs across three random seeds from CoRNStack, the original Stack v2, and other contrastive code training datasets like CosQAand CodeSearchNet (CSN). The results indicate that CoRNStack has a significantly higher mean percentage of correct pairings. We further provide below the language-wise correctness for the mean numbers reported in the paper and add this table to section A.3 of the revised version.
>
> | Dataset            | Python | Java  | Javascript | PHP   | Go    | Ruby  | Avg  |
> |--------------------|--------|-------|------------|-------|-------|-------|------|
> | Stack v2          | 54.8   | 50.3  | 53.6       | 56.4  | 63.2  | 39.2  | 52.9 |
> | CSN               | 53.9   | 56.3  | 50.5       | 60.7  | 65.9  | 47.6  | 55.8 |
> | CosQA             | 63.9   | -     | -          | -     | -     | -     | 63.9 |
> | **Ours (CoRNStack)**  | **76.2**   | **80.6**  | **74.4**       | **77.3**  | **82.8**  | **71.4**  | **77.1** |
>
> ***Question 2: It should also be clarified whether the code data in the dataset has been checked manually or tested with examples, as this is important for the quality of the code retrieval tasks.***
>
> **Response:**
>
> We acknowledge that the reviewer raises a very interesting point. With CoRNStack being a large-scale training dataset containing millions of data points, any validation must be automated, since it is not feasible to manually check the quality of each code snippet.
>
> To ensure the syntactic correctness of the code data in CoRNStack, we followed the approach in [1] by using the Tree-sitter parsing toolkit to filter out any functions that cannot be parsed into a syntax tree during data extraction from The Stack v2.
>
> For evaluating the functional correctness, we employed a heuristic validation method described in our response to Question 1. Specifically, we utilized a code language model to judge whether a given positive code snippet fully answers its corresponding query. Our results, shown in Table 2 of the paper, demonstrate that CoRNStack has a significantly higher mean percentage correctness than existing code datasets.
>
> Another way of validating the functional correctness of the code data would be to generate test cases using a combination of LLMs and/or human annotators.  However, considering the scale of the dataset, this approach would be computationally intensive to generate diverse test cases that cover all edge cases and would require substantial resources to set up the necessary environments for execution-based validation. Nevertheless, we agree that this is a valuable area for future exploration and include a discussion of this in lines 500-505 of the revised paper.
>
> [1] “GraphCodeBert: Pretraining Code Representations with Data Flow”; Guo et al, ICLR 2021

---

> > ### Author Response · Authors · 2024-11-19
> > **Response to Reviewer MeoD (2/2)**
> >
> > ***Question 3: Additionally, it would be helpful to include information about the variety of code tasks in the new dataset, particularly how simple and complex scenarios are distributed.***
> >
> > **Response:**
> >
> > CoRNStack comprises (docstring, function) instances in the form of (text, code) pairs, following previous contrastive code datasets such as CosQA [1] and CodeSearchNet (CSN) [2]. We chose this format due to the abundance of such pairs available on open platforms like GitHub, which allows us to create a large-scale training dataset.
> >
> > Despite CoRNStack containing only text-code pairs, finetuning our code retriever on this dataset yields state-of-the-art performance (Table 4 of the paper) on the CoIR benchmark [3], which includes diverse code retrieval tasks such as text-to-code retrieval, code-to-text retrieval, code-to-code retrieval, and hybrid code retrieval (retrieving a combination of code and textual documents given a hybrid query). The superior cross-task generalization of our code retriever demonstrates that CoRNStack effectively captures the underlying relationships and semantics necessary for a wide range of code retrieval tasks.
> >
> > CoRNStack is derived from The Stack v2 [4], which aggregates multiple sources such as the Software Heritage source code archive, GitHub pull requests, Stack Overflow, Kaggle notebooks, and code documentation websites. This diverse range of sources ensures our dataset encompasses a wide variety of coding scenarios and domains.
> >
> > To provide more insight into the variety and complexity of code tasks in CoRNStack, we analyzed the distribution of code topics for 100k randomly sampled instances using Nomic Atlas, a popular unstructured text visualization tool. Nomic Atlas employs a cluster-based keyword identification algorithm and leverages a language model to generate topics. We find that the majority of examples fall into eight broad categories: object creation, data sorting, data management, API management, configuration, data validation, graphics, and math operations. We have added a word cloud in Section A.2 of the Appendix to illustrate popular fine-grained topics within these categories. Additionally, the interactive Nomic Atlas map of 100k randomly sampled instances from CoRNStack is available here: [link](https://atlas.nomic.ai/data/corniclr25/cornstack-100k/map)
> >
> > We hope this additional information addresses the reviewer’s concern, and we ensure it is included in the revised version of the paper.
> >
> > [1] “Cosqa: 20,000+ web queries for code search and question answering”; Huang et al, 2021
> >
> > [2] “Codesearchnet challenge: Evaluating the state of semantic code search”; Husein et al, 2019
> >
> > [3] “CoIR: A Comprehensive Benchmark for Code Information Retrieval Models”; Li et al, arXiv 2024
> >
> > [4] “StarCoder 2 and The Stack v2: The Next Generation”; Lozhkov et al, arXiv 2024

---

> ### Author Response · Authors · 2024-11-25
> **Following Up on Review**
>
> Hello Reviewer MeoD,
>
> We would like to thank you for the thoughtful and constructive feedback on our submission.
>
> As the rebuttal period comes to a close, we are following up to inquire if our responses have addressed the concerns and would appreciate **you considering raising your final rating** or providing further feedback/concerns. We are looking forward to your response!
>
> Thanks,
> ICLR 2025 Conference Paper 12599 Authors

---

> > ### Comment · Reviewer_MeoD · 2024-11-27
> >
> > Thank you for your responses. I will keep my score.

---

> > > ### Author Response · Authors · 2024-12-02
> > >
> > > We thank the reviewer for their positive score and valuable feedback.

---

### Official Review · Reviewer_g16n · 2024-11-04

**Soundness:** 3
**Presentation:** 3
**Contribution:** 3
**Rating:** 8
**Confidence:** 4

**Summary:**

This paper proposes a new dataset CoRNSTACK for code retrieval tasks. Different from previous datasets, this dataset is a large-scale and high-quality dataset that contains hard negative samples. Trained with CoRNSTACK, the performance of models on code retrieval tasks and code reranking tasks has significantly increased. Besides, the authors also demonstrate the effectiveness of training with the proposed dataset in a real-world application (addressing GitHub issues).

**Strengths:**

1. According to my knowledge, the problems in previous code retrieval datasets discussed by the authors truly exist and the proposed dataset addresses these problems well. This proposed high-quality dataset will benefit research in code retrieval tasks and real-world retrieval applications.
2. The idea of fine-tuning LLMs as code rerankers is interesting and effective.
3. The authors not only evaluate code retrieval tasks, but also demonstrate the effectiveness of training with CoRNSTACK in addressing real-world GitHub issues.

**Weaknesses:**

Overall, I think this paper is good and should be accepted. All of the following weaknesses are minor problems.
1. There is no analysis of the reason for the improved performance of code rerankers. I guess this is also due to the hard negative samples?
2. As stated by the authors, one of the motivations for constructing this dataset is that existing datasets contain many false negative samples. In [1], the authors also discussed this problem and proposed methods to address the problem during model fine-tuning, which I think should also be discussed in the paper.
3. Although the authors discussed the problems of existing datasets in the introduction and some preliminaries of code retrieval in the paper, I still think it is better to write a related work section to make them clearer. Since code retrieval is not an extremely hot topic, many readers may not know many details of previous works in this area.
4. Typo in Line 272, 'high-qualit,y'


[1] "Rethinking Negative Pairs in Code Search." Proceedings of the 2023 Conference on Empirical Methods in Natural Language Processing. 2023.

**Questions:**

1. Why do you use pre-trained text embedding models for dual consistency filtering? What about using pre-trained code retrieval models pre-trained on previous code retrieval datasets?
2. Fine-tuning models with contrastive learning is a common and traditional method for code retrieval tasks. Recently, there has been a new paradigm named Generation-Augmented Retrieval (GAR) framework proposed for code retrieval tasks as well [1, 2]. I think this framework makes the fine-tuning of code retrieval models somewhat unnecessary. What's your opinion on the necessity of fine-tuning code retrieval models? (Also, I think discuss this point in the paper will be better)


[1] Haochen Li, Xin Zhou, and Zhiqi Shen. 2024. Rewriting the Code: A Simple Method for Large Language Model Augmented Code Search. In Proceedings of the 62nd Annual Meeting of the Association for Computational Linguistics (Volume 1: Long Papers), pages 1371–1389, Bangkok, Thailand. Association for Computational Linguistics.

[2] "Generation-Augmented Query Expansion For Code Retrieval." arXiv preprint arXiv:2212.10692 (2022).

---

> ### Author Response · Authors · 2024-11-24
> **Response to Reviewer g16n**
>
> We would like to thank the reviewer for appreciating the paper with a positive score. Here, we address the reviewer’s concerns and questions:
>
> ***Weakness 1: There is no analysis of the reason for the improved performance of code rerankers. I guess this is also due to the hard negative samples?***
>
> **Response**: We attribute the improvement in performance to the finetuning of LLMs for the listwise code reranking task. As Table 6 in the paper shows, the instruction-tuned code generation model, when directly used for listwise code reranking, has considerably lower performance. However, it improves after being finetuned for this task, using the data made possible by getting a relevance ordering on the negatives present within CoRNStack.
>
> ***Weakness 2: As stated by the authors, one of the motivations for constructing this dataset is that existing datasets contain many false negative samples. In [1], the authors also discussed this problem and proposed methods to address the problem during model fine-tuning, which I think should also be discussed in the paper.***
>
> **Response**: We sincerely thank the reviewer for highlighting this relevant work. The paper mentioned introduces the Soft-InfoNCE loss to reduce the impact of false negatives with contrastive training when using InfoNCE loss, by weighting negative samples based on their similarity to the query. We view this work as complementary to ours, as the mined hard negatives from CoRNStack could be seamlessly integrated into the Soft-InfoNCE loss, in a similar fashion to that described in lines 236–248 of our paper, to further improve effectiveness. We appreciate the reviewer bringing up this work and will incorporate a discussion into the final version.
>
> ***Weakness 3: Although the authors discussed the problems of existing datasets in the introduction and some preliminaries of code retrieval in the paper, I still think it is better to write a related work section to make them clearer. Since code retrieval is not an extremely hot topic, many readers may not know many details of previous works in this area.***
>
> **Response**: We thank the reviewer for this suggestion. We are actively working on drafting a related work section and will add it into the revised paper by the camera-ready deadline at the latest.
>
> ***Question 1: Why do you use pre-trained text embedding models for dual consistency filtering? What about using pre-trained code retrieval models pre-trained on previous code retrieval datasets?***
>
> **Response**: We use the pre-trained code embedding model Jina-Code-v2 as the proxy embedding model N (line 166 in the paper) for dual consistency filtering. We selected this embedding model for filtering because of its performance on code retrieval benchmarks like CodeSearchNet, AdvTest, and CoIR. We have clarified this detail in footnote 4 of the revised paper.
>
>
> ***Question 2: Fine-tuning models with contrastive learning is a common and traditional method for code retrieval tasks. Recently, there has been a new paradigm named Generation-Augmented Retrieval (GAR) framework proposed for code retrieval tasks as well [1, 2]. I think this framework makes the fine-tuning of code retrieval models somewhat unnecessary. What's your opinion on the necessity of fine-tuning code retrieval models?***
>
> **Response**: We thank the reviewer for highlighting the Generation-Augmented Retrieval (GAR) framework. We acknowledge that GAR introduces an innovative approach by using a generation model to expand queries with exemplar code snippets, addressing issues of short or ambiguous queries. However, the retrieval component in GAR still depends on a model trained with contrastive learning to match the expanded queries with relevant code snippets effectively. Therefore, we believe that fine-tuning code retrieval models remains essential to enhance their ability to understand code semantics and improve retrieval accuracy. Additionally, GAR may introduce additional latency due to the generation step, whereas using finetuned retrieval models directly provides faster results. In summary, finetuning code retrieval models is still necessary for optimal performance, even when employing GAR techniques.

---

> > ### Comment · Reviewer_g16n · 2024-11-25
> > **Resposne to Authors**
> >
> > I would like to thank the authors for their clarification. I will keep my score.

---

> > > ### Author Response · Authors · 2024-12-02
> > >
> > > We thank the reviewer for their positive score and thoughtful comments. We are glad that the reviewer appreciates the significance of our proposed dataset in addressing the limitations of existing code retrieval datasets, the effectiveness of our approach of fine-tuning LLMs as code re-rankers, and how our dataset will benefit future research in code retrieval tasks and real-world retrieval applications.

---

### Meta-Review · Area_Chair_7aZk · 2024-12-21

**Metareview:**

This paper introduces CoRNStack, a large-scale, high-quality dataset designed to improve code retrieval models for practical applications. The reviewers appreciate the innovative design of the dataset and its demonstrated potential to enhance code retrieval tasks. However, there are some minor concerns regarding the clarity of the dataset's details and sources. The authors are encouraged to better explain the process of creating the dataset and to broaden the experimental evaluation.

**Additional Comments On Reviewer Discussion:**

The authors have effectively addressed most of the reviewers' comments during the rebuttal stage.

---

### Decision · Program_Chairs · 2025-01-22

Accept (Poster)